# Parental Factors Associated with Child or Adolescent Medication Adherence: A Systematic Review

**DOI:** 10.3390/healthcare11040501

**Published:** 2023-02-08

**Authors:** Clarisse Roswini Kalaman, Norhayati Ibrahim, Vinorra Shaker, Choy Qing Cham, Meng Chuan Ho, Uma Visvalingam, Farah Ahmad Shahabuddin, Fairuz Nazri Abd Rahman, Mohd Radzi Tarmizi A Halim, Manveen Kaur, Fatin Liyana Azhar, Amira Najiha Yahya, Rohana Sham, Ching Sin Siau, Kai Wei Lee

**Affiliations:** 1Center for Healthy Ageing and Wellness (H-CARE), Faculty of Health Sciences, Universiti Kebangsaan Malaysia, Kuala Lumpur 50300, Malaysia; 2Institute of Islam Hadhari, Universiti Kebangsaan Malaysia, Bangi 43600, Malaysia; 3School of Psychology, Asia Pacific University of Technology and Innovation, Kuala Lumpur 57000, Malaysia; 4Center for Pre-U Studies, UCSI University, Cheras 56000, Malaysia; 5Department of Psychiatry, Hospital Putrajaya, Putrajaya 62250, Malaysia; 6Department of Psychiatry, Hospital Bahagia Ulu Kinta, Tanjung Rambutan 31250, Malaysia; 7Psychiatry Department, Faculty of Medicine, Universiti Kebangsaan Malaysia, Kuala Lumpur 56000, Malaysia; 8Faculty of Business, Economics and Human Development, University Malaysia Terengganu, Kuala Terengganu 21030, Malaysia; 9Department of Psychological Medicine, Faculty of Medicine, University Malaya, Kuala Lumpur 50603, Malaysia; 10Department of Educational Psychology & Counselling, Faculty of Education, Universiti Malaya, Kuala Lumpur 50603, Malaysia; 11School of Business, Asia Pacific University of Technology and Innovation, Kuala Lumpur 57000, Malaysia; 12Center for Community Health Studies (ReaCH), Faculty of Health Sciences, Universiti Kebangsaan Malaysia, Kuala Lumpur 50300, Malaysia; 13Department of Medical Microbiology, Faculty of Medicine and Health Sciences, Universiti Putra Malaysia, Serdang 43400, Malaysia

**Keywords:** parenting, medication adherence, children and adolescent, mental illness

## Abstract

Medication adherence, especially among children and adolescents with psychiatric disorders, is often seen as a major treatment challenge. The purpose of this study is to systematically review studies addressing specific aspects of parental factors that are positively or negatively associated with medication adherence among children and adolescents with psychiatric disorders. A systematic literature search of English language publications, from inception through December 2021, was conducted from PubMed, Scopus, and MEDLINE databases. This review has complied with Preferred Reporting Items for Systematic Reviews and Meta-Analyses statement guidelines. A total of 23 studies (77,188 participants) met inclusion criteria. Nonadherence rates ranged between 8% to 69%. Parents’ socioeconomic background, family living status and functioning, parents’ perception and attitude towards the importance of medication taking in treating psychiatric disorders, and parents’ mental health status are significant parental characteristics associated with medication adherence in children and adolescents with psychiatric disorders. In conclusion, by identifying specific parental characteristics related to the medication adherence of children and adolescents with psychiatric disorders, targeted interventions on parents could be developed to guide parents in improving their child’s medication adherence.

## 1. Introduction 

Mental disorders in children and adolescents are common and possess a significant impact on their well-being in the long-run [1]. There are a number of reviews that have reported the significant increase in prevalence of mental disorders among children and adolescents, especially after the COVID-19 pandemic outbreak [2,3,4]. Globally, an estimated 13% of adolescents aged 10–19 years old [5] experienced mental disorders that are often times unrecognized and untreated [6,7]. Some of the most commonly diagnosed mental disorders among children and adolescents are Attention-Deficit/Hyperactivity Disorder (ADHD), behavioral problems, depression, and anxiety [1,7]. Since there are no definitions that established the exact boundaries of the psychiatric disorder concept, hence, as stated in the Diagnostic and Statistical Manual of Mental Disorders, Fifth Edition, Text Revision (DSM-5-TR), mental disorders, also known as psychiatric disorders, may be conceptualized as a “*clinically significant behavioral or psychological syndrome or pattern, associated with present distress or disability, and is not merely an expected response to common stressors and losses or a culturally sanctioned response to a particular event, instead was primarily a result of social deviance or conflicts with society, that occurs in an individual*” [8] (p. 5). According to the WHO [7], failure to address and deal with adolescents onset mental health conditions may lead to suicide, which is one of the most devastating causes of death, especially among 15- to 19-year-olds. Adolescence is a transitional phase to adulthood that is marked by various biological, cognitive, and psychosocial changes [9]. Hence, it is a crucial period for family, educators, and the community, through various mental health interventions, to promote mental health among adolescents. However, one of the major challenges often faced by psychiatrists in promoting mental well-being and preventing mental health deterioration, especially among children and adolescents diagnosed with mental disorders, is medical adherence towards prescribed treatments and medication [10,11]. 

According to the WHO [12], medical adherence can be defined as the extent of an individual’s efforts or behavior in observing treatment-related instructions such as taking medication, following a recommended diet, modifying habits, attending treatment appointments, abiding to medication prescription, and corresponding with agreed recommendations from a healthcare professional. Non-adherence is seen as one of the major obstacles and common causes to the increase of mental illness relapses, hospitalization rates, morbidity, and other harmful outcomes [10,11,13,14,15,16]. In reviews conducted among adolescents with psychiatric disorders who were prescribed psychotropic medications or received treatment from psychiatric services, 28% to 75% prematurely dropped out of treatment, a median of 33% adolescents were medically non-adherent, 44% reported no reliable change, 6% reported reliable deterioration, and 13.2% were often re-hospitalized due to suicidal attempts made after discharge within a year [15,17,18,19]. There were not many studies conducted on medical adherence among children, ages 3 to 12 years old, with psychiatric disorder. Findings from Edgcomb et al. [19] suggest that in comparison with adolescents, children tend to have a higher likelihood of reporting adherence to medication. However, according to Edgcomb et al. [19], review on children and adolescents with psychiatric disorder, medical nonadherence is a widespread problem and should be provided with equal importance as children and adolescents with other chronic medical illnesses. 

There are many determinants of medication adherence that were reported and categorized in various ways [11,14,16,20,21,22]. The WHO [12] suggested five main categories that covers the multidimensional phenomenon of adherence, which are: (1) socio-economic factors (low socioeconomic status, illiteracy, lack of family support); (2) provider-patient/health care system factors (poor medication distribution, therapeutic relationship); (3) therapy-related factors (complexity of medical regimens, duration of treatments or the immediacy of beneficial effects); (4) condition-related factors (severity of symptoms, rates of progression or level of disability); and (5) patient-related factors (knowledge and beliefs, self-determination). Clinical outcomes pertaining to adherence may be common across all branches of medicine; however, non-adherence among psychiatric patients, in comparison with patients receiving medication or treatment for physical conditions, poses additional challenges, such as having to deal with suicidal ideation and emotional outbursts, relapses as well as stigmatizing attitudes from the patient and the public, that increases the risks of morbidity [13]. 

Medication adherence, defined as the degree to which patients’ medication-taking behavior corresponds with the agreed, prescribed medication dosing regimen provided by a healthcare professional, is an important subset of the broad study of medical or treatment adherence [12]. The lack of medication adherence poses a significant impact in increasing the risk of psychiatric disorder recurrence and suicidality in adulthood [7,23,24]. Since children and adolescents are still under the purview of parents or parental caregivers, hence adherence among the younger age psychiatric patients are often times largely dependent on the ability of the parent or parental caregiver to understand and follow through with prescribed medication regimens [12]. 

Parental influence on children and adolescents’ well-being has been widely investigated. Parental influence is inclusive of all influences related to the paternal and maternal figure, that affects the physical, emotional, and intellectual development of a child [25]. According to a review conducted by Rohden et al. [26], parental factors such as income, age characteristics, family structure, parents’ well-being, parental care or neglect and parental arbitration on child’s adherence towards treatment and medication are salient aspects and can be seen as a risk or protective factor of a child’s well-being. Children or adolescents with a psychiatric disorder may possess limited knowledge on mental health and lack of ability in accessing the Child and Adolescent Mental Health Services (CAMHS), thus requiring parental supervision and guidance in overcoming structural barriers such as financial costs and logistical barriers, as well as adherence challenges such as monitoring symptom severity and ensuring the medication is administered appropriately [27]. Overall, parental factors can be seen as a crucial factor in maximizing good clinical outcomes or causing a major health setback. 

There are few reviews that have reported the role of parents as one of the factors associated with medication adherence among children and adolescents with psychiatric disorder [10,15,16]. Both Edgcomb and Zima [10] and Häge et al. [15] investigated predictors of medication adherence only, while Timlin et al. [16] reviewed factors associated with adolescents’ adherence to both medication and non-pharmacological treatments in mental health. A total of 60 studies were reviewed in Edgcomb and Zima [10], Häge et al. [15], and Timlin et al. [16]. The reviews concluded that the range of medication nonadherence was wide, between 6% and 62%, and was considered a common problem in mental health care among children and adolescents with a psychiatric disorder. Factors such as illness severity, comorbidity burden or underlying diagnosis, substance use, and attention-deficit/hyperactivity disorder, age, sex, interpersonal care processes and the adolescent’s own beliefs towards treatment emerged as significant predictors of adherence. With regard to parental factors, the findings from these reviews suggests that positive attitudes or the level of support obtained from family members were associated with higher adherence among children and adolescents with psychiatric disorders [10,15,16]. Nevertheless, Timlin et al. [16] pointed out the fact that it is challenging to ensure adolescents’ medication adherence to prescribed treatment or medication regimens, as they are transitioning into adulthood and tend to become more independent of their parents. However, these reviews did not provide a clear synthesis of literature that highlights specific components of the parental factors associated with child/adolescent medication adherence. Häge et al. [15] also emphasized the need for future research that involves familial factors associated with medication adherence among children and adolescents with psychiatric disorders. Thus, the purpose of this review is to evaluate the peer-reviewed literature addressing specific aspects of parental factors that are positively or negatively associated with medication adherence among children and adolescents diagnosed with psychiatric disorders. 

The specific questions addressed in this review were: How was medication adherence and/or non-adherence among children and adolescents with psychiatric disorders defined?What are the parental characteristics associated with medication adherence among children and adolescents with psychiatric disorders?

## 2. Materials and Method

### 2.1. Protocol

This review was performed according to Preferred Reporting Items for Systematic Reviews and Meta-Analyses (PRISMA) guidelines and was registered with PROSPERO (CRD42021256211).

### 2.2. Search Strategy

A search of articles published relevant to parenting and medical adherence among children and adolescents with psychiatric disorders was conducted. The systematic search of English language publications, from inception through December 2021 was conducted using three main electronic databases: PubMed, Scopus, and MEDLINE. As shown in Table 1, searches were piloted, and as a result, a range of search terms with broader descriptions of parenting, medical adherence and psychiatric disorders were tailored to meet specific requirements of each database. All searches were placed within titles and abstracts to maximize the yield of a large data, ensuring as wide as possible a coverage in the review. In addition, through chain searching, reference lists of systematic reviews conducted by Edgcomb and Zima [10], Häge et al. [15], and Timlin et al. [16] were screened and eligible articles were included in this review. 

### 2.3. Inclusion and Exclusion Criteria

The systematic searches were designed to identify studies that investigate the relationship of parental factors and child/adolescent medication adherence, targeting all children and adolescents, ages ranging from 1 to 19 years, who were prescribed medication for psychiatric conditions. The present review included studies that (1) were quantitative; (2) discussed any form of parental factor; (3) analyzed medication adherence as the outcome variable (i.e., adherence toward observing instructions on taking prescribed medications); and (4) were published in English or possessed English translations. Since this review is confined to including quantitative study design term only, hence pilot, validation, psychometric, preliminary, systematic reviews, meta-analysis, qualitative, randomized-controlled trial, interventional, and treatment-related studies were excluded from this review, increasing the robustness of findings derived from this review. Articles that discussed medical adherence among children and adolescents with psychiatric disorders yet without including any parental factor, and vice versa, were also excluded from the review. 

### 2.4. Study Selection 

A PRISMA flowchart documenting the process of study selection is shown in Figure 1. After the removal of duplicate publications using the Endnote Program X5 software, the study selection process was screened by two reviewers in three stages. All potential articles identified for inclusion, from eligibility assessment of title and abstract, were independently assessed. If the reviewers coded an article as potentially eligible, the full-texts were then retrieved and reviewed to confirm eligibility. Articles excluded at every stage are agreed to have met at least one of the exclusion criteria outlined. Any disagreement between the reviewers were discussed with a third reviewer until a consensus was reached. 

### 2.5. Data Extraction

Data extraction was performed using a structured data collection sheet developed using the Microsoft Excel software and was piloted beforehand. As shown in Table 2, extracted data includes: (1) study identification features such as authors, year of publication; (2) study characteristics such as study design; and (3) population characteristics and sample size. Data was extracted by one reviewer (CRK) while the second reviewer (CQC) verified the completeness and accuracy of the extracted data. All available relevant data was extracted from the reviews and no additional information was sought from the authors. 

### 2.6. Quality Assessment

The quality of the paper included was assessed using the “Strengthening the Reporting of Observational Studies in Epidemiology (STROBE)” checklist by von Elm et al., [51]. There are 22 proposed items in the checklist, with items number 6, 12, 14, and 15 having specific variations that assessed 6 components for cohort, case–control and cross-sectional studies. The absence or presence of component stated in each item from the article will be graded with a “0” or a “1”, respectively. A total STROBE score of ≥14/22 assessed for each article are graded as ‘low risk bias’, while articles with a total STROBE score of <14/22 are graded as ‘high risk bias’. The results of the study quality assessment are shown in Appendix A, where 16 of the articles were rated to be at low risk of bias while the remaining 7 were rated to be at high risk of bias. Common reasons for loss of points in articles were: lack of reporting on potential sources of bias; not addressing the handling of missing data; lacking sample size justifications; insufficient description of statistical analyses; and not reporting of effect sizes, confidence intervals, and funding details. 

### 2.7. Statistical Analysis

Due to the differing data sources, heterogeneity and/or the small number of studies included in this review, a quantitative analysis was considered inappropriate and unsuitable. Instead, a narrative overview of the data from included studies (e.g., study characteristics, participants, outcomes, and findings) were presented with tabular summaries for an overall description in this review. The data were synthesized by categorizing the components of parental factors and psychiatric disorders the studies examined. Medication adherence outcomes were extracted and used as the main findings for this review. All data in the table were harmonized so that the influence on adherence refers to an increase in the factor regardless of whether the factor is positive (i.e., associated with higher medication adherence) or negative (i.e., associated with lower medication adherence). 

## 3. Results 

In total, 3006 articles pertaining to both parental factors and medication adherence factors among children and adolescents with psychiatric disorders were identified from three databases: Scopus (413), PubMed (2044), MEDLINE (549), and chain searching (60). Then, a total of 2531 unduplicated articles were further screened through title and abstracts. Specifically, 532 articles were duplicates and 2265 articles were irrelevant. The remaining 40 eligible articles, after title and abstract screening, were reviewed in their entirety, resulting in the further removal of 17 articles that failed to meet all necessary criteria: parental factors were not thoroughly discussed [52,53,54,55,56,57,58]; target population was not the child or adolescent population with psychiatric disorders [59,60]; outcome measures were not focused on medical adherence [61,62,63,64]; and medication adherence was not measured based on administering psychotropic medications [65,66,67,68]. Finally, only 23 articles were included in this review. 

### 3.1. Summary of Study Characteristics

A total of 77,188 children and adolescents with psychiatric disorders and whom were prescribed psychotropic medication were included in this review (Table 2). A majority of 15 studies was conducted in the United States of America. There were two studies conducted in Canada [38,48], and one study each conducted in Italy [28], Turkey [29], Australia [34], the United Kingdom [41], Mexico [46], and Finland [49]. Even though Harpur et al. [41] conducted the study in the United Kingdom, however participants from Canada, Germany, Australia, Israel, Singapore, Republic of Ireland, South Africa, Brazil, and Malaysia participated in the study through the internet. 

Participants were largely recruited from clinical settings, such as the inpatient (*n* = 10) and outpatient (*n* = 8) clinical psychiatric facilities. Harpur et al. [41] and Pérez-Garza et al. [46] recruited participants from clinics, yet specific services in which participants were treated were not mentioned. Additionally, Harpur et al. [41] also recruited participants in the community though parent support groups and the internet. Similarly, Demidovich et al. [36] recruited participants through newspapers, radio advertisements, and brochures sent to schools and local mental health centers, as well as program sites affiliated with the University of Pittsburgh Medical Center. On the other hand, Bushnell et al. [32] utilized enrollment files, inpatient and outpatient services, and dispensed prescriptions obtained from the MarketScan Commercial Claims database to identify and recruit participants. Three studies recruited participants from a larger project or separate study [42,44,50]. 

Almost all studies included in this review (*n* = 22) possess participants from the adolescent age group. A total of nine studies recruited both children and adolescent participants, with a minimum age of 3 years and a maximum of 18 years. Among the included studies, Demidovich et al. [36] was the only study conducted among children only, with ages ranging between 6 to 11 years old. The included studies employed naturalistic (*n* = 4), prospective (*n* = 3), cross-sectional (*n* = 2), exploratory (*n* = 1), retrospective (*n* = 1), and mixed-method (*n* = 1) study designs. The remaining eleven studies did not mention the study design. The study duration was mentioned in nearly half of the studies (*n* = 20) included in this review, and they were from a minimum of 3 weeks to a maximum of 3 years. 

A total of 18 studies clearly reported the prevalence of medication adherence/nonadherence among the children/adolescents [28,29,31,32,33,34,35,36,38,39,40,43,44,45,47,48,49,50]. The proportion of partial or complete medication adherence ranged from 27% to 78%, while the proportion of medication nonadherence ranged from 8% to 69%.

### 3.2. Parental Factors Associated with Child/Adolescent Medication Adherence

Detailed information on the association of parental factors and offspring medication adherence is presented in Table 3. 

A few studies investigated the association between family socioeconomic status with medication adherence among children and adolescents (*n* = 5). Socioeconomic status was found to be positively associated with medication adherence in DelBello et al. [35], Demidovich et al. [36], and Harpur et al. [41]. Bernstein et al. [30] on the other hand reported no significant association between socioeconomic status and medication adherence. Another sociodemographic factor studied in conjunction with children and adolescents medication adherence was parental education (*n* = 3). It was found that there was no statistically significant relationship between parental education level and medication persistence or commitment [29,37]. Even though Moses [44] reported a positive correlation (*p* < 0.05) between parents’ education and youth’s commitment to medication, when multiple logistic regression was utilized to evaluate the predictive value of the significant correlates, parent education then demonstrated a non-significant statistical trend (*p* = 0.08). 

Besides that, several studies investigated the influence of family living status on medication adherence and found that there was no significant association [39,40,43,44,48]. In contrast, Atzori et al. [28] stated that youths with poor family structure or are not living with both parents led to a lack of psychotherapy or educational resources to cope with their mental health condition, which in turn made medication the more accessible treatment in comparison to long-term therapy sessions. 

Apart from family living status, the role of family functioning and relationship on child or adolescent medication adherence were also investigated. In this regard, family functioning or parental involvement in the child’s medication routine is a strong predictor studied in many articles related to child/adolescent medication adherence (*n* = 11). All studies included in this review that investigated family functioning and relationship reported that dysfunctional families, with the least affectionate parent–child relationship predicting low medication adherence [30,37,42,43,44,48,49]. Regarding family support, there were several studies that reported non-statistically significant relationship between family support and child/adolescent medication adherence [36,38,40]. In a study conducted by Gearing et al. [38], increased social support was not associated with improved adherence due to the precipitated bias reported from excluding participants with high levels of family support in the study. 

Parents’ psychological health, substance use, history of psychotropic medication or psychiatric disorders and lifetime history of parental hospitalization were factors studied in association with their child’s medication adherence (*n* = 7). According to Burns et al. [31] and Gearing et al. [38], current psychopathology of parents was associated with lower medication adherence of their child, whilst a history of psychopathology was not significantly associated. Similarly, Drotar et al. [37] stated that lifetime history of maternal (*r* = −0.31; *p* < 0.01) and paternal (*r* = −0.44; *p* < 0.01) hospitalization for psychiatric illness was associated with their child’s medication nonadherence. According to King et al. [43] the mother’s depressive, paranoid, and hostile symptoms were associated with worse medication follow-through of the child. 

Similarly, Bushnell et al. [32] also reported that among the number of psychiatric diagnoses that were evaluated in either parent of each child, parent substance use disorder diagnosis was identified as an independent predictor of child’s adherence. However, in contrast, Demidovich et al. [36] and Timlin et al. [49] stated that there was no significant connection observed between parent’s psychiatric problems or substance use towards the child’s medication adherence. Despite having observed a significant association between mother’s depressive, paranoid and hostility symptoms towards the child’s medication adherence, King et al. [43] did not fail to also report that mother’s anxiety symptoms and father’s psychopathology were seen to be non-significant to the child’s medication adherence. According to Bushnell et al. [32], parents’ psychological conditions may be seen as a significant predictor to child/adolescent nonadherence. However, parents with psychiatric disorders who partake in preventative measures or healthy medication adherence behaviors, such as taking daily medications, making trips to the pharmacy, and parents well/preventative visits, encourage adherence in the child [32]. Consequently, parental behavior or attitudes toward psychotropic medication is another major factor which was studied in conjunction with child/adolescent medication adherence (*n* = 8). In a study conducted by Coletti et al. [33], it was reported that more information is needed to confirm the association between perceived effectiveness and psychiatric condition medication adherence. However, over the years, many studies have reported a significantly positive association between parent’s perceived efficacy and acceptability of psychotropic medication towards the child’s medication persistence or adherence [29,31,41]. Parental self-efficacy, resiliency, emotional support for the child, stigma, perceived costs of medication and parent request of medication discontinuity are of the several parental behaviors that have been reported to decrease the likelihood of the child’s medication adherence [36,41,47].

### 3.3. Definition of Medication Adherence and Nonadherence

The definition of medication adherence varied across studies, depending on the prescription of medication and clinical outcome. However, the definition of nonadherence, as discontinuation or termination of medication at any given time, seems to be a commonly used definition in all studies. In a study conducted by Moses [44], medication adherence is seen as an expression of commitment, hence the terms used to classify adherent and non-adherent youths in this study are “committed” and “less committed”. In studies conducted by Ayaz et al. [29], Bushnell et al. [32], and Demidovich et al. [36], “medication acceptors/medication persistence” or “medication refusers/discontinuation” were some of the synonymic terms used to address adherence, however the term “compliance/adherence” or “noncompliance/nonadherence” were among the common terms often used interchangeably (cf. Table 4). Majority of the studies (*n* = 20) included in this review mainly attempted to investigate medication adherence among children or adolescents with psychiatric disorders. However, there are three studies of which objectives were not focused on investigating medication adherence, nonetheless were included in this review. Demidovich et al. [36] reported significant effects of parental medication acceptability and a child’s decision to accept or refuse medication recommendation after the administration of a modular psychosocial treatment, hence were included in this review. Similarly, Hoza et al. [42] emphasizes the primary role of parents as implementers of treatment, indirectly predicting the success or failure of children treatment outcomes. A study conducted by Harpur et al. [41] was mainly focused on describing the psychometric properties of the Southampton ADHD Medication Behavior and Attitudes (SAMBA) scale. Nevertheless, this article was still included in this review due to the reliable and valid function of the scale in measuring parental stigma that significantly predicts pediatric medication adherence.

All articles included in this review conducted quantitative methods in measuring medication adherence of children and adolescents with psychiatric disorder. Medication adherence was assessed through questionnaires or scales in eleven studies [29,31,33,36,38,40,41,42,45,46,48], structured or semi-structured interviews in six studies [29,34,39,43,44,47], manual or electronic pill counts in five studies [28,30,37,40,50], clinical measurements such as blood levels, serum concentration, etc. in three studies [30,32,37,50], and hospital medical records in two studies [29,35,44,49]. In fourteen of the studies, adherence among children or adolescents with psychiatric disorders was assessed on the basis of questionnaires or interviews with the parents, caregivers, or physicians. In a study conducted by Dean et al. [34], medication adherence was reported primarily by the child and verified by the parents’ report of their child’s medication adherence through an open-ended question on parental involvement in medication monitoring. Similarly, instead of solely relying on self-report medication adherence from patients, Goldstein et al. [40] and Moses [44] also referred to Appendix A, such as pill count and medical records, as part of an objective form in measuring medication adherence. Studies conducted by Burns et al. [31], Harpur et al. [41], and Munson et al. [45] correlated the scores to ensure an agreement is reached between parent and child report of adherence. Likewise, Pogge et al. [47] reviewed and corroborated adherence assessment with additional informant when patient’s assessments appeared unreliable.

## 4. Discussion

The main objective of this review was to summarize existing evidence of the associations between parental factors and medical adherence among children and adolescents with psychiatric disorders. This study also sought to summarize the definition of medication adherence as employed in the reviewed studies. The overall findings from the 23 studies included in this review reflect the ubiquitous impact parents have in effects to the child’s medication taking behaviors. Since the number of included studies was low and the quality of evidence varied across studies, the review only allows a narrow look at the various factors of medication adherence among children and adolescents with psychiatric disorders. However, in comparison with three other systematic reviews conducted to explore general factors of medical adherence among children and adolescents with psychiatric disorders, this review reports the results of a thorough investigation on parental factors associated with children and adolescents medication adherence. Parental factors were described in various aspects, the most common being parent’s socioeconomic background, family living status and functioning, parent’s perception and attitude towards the importance of medication taking in treating psychiatric disorders, and parental mental health status.

Most studies that investigated the relationship between socioeconomic status and child/adolescent medication adherence showed a significant positive relationship between the variables [35,36,41]. Socioeconomic background has been consistently associated with disparities in child and adolescent mental health [69,70]. This review further confirmed that socioeconomic status may have contributed to mental health disparities through medication adherence. Families with low socioeconomic status may perceive the costs of mental health medication and treatment as an additional burden instead of a need, hence leading to higher levels of nonadherence [35,36,41]. However, reasons were unknown as to why lower and lower-middle socioeconomic status families in the study conducted by King et al. [43] reported higher rates of complete medication follow-through compared to other families.

The findings of this review emphasizes the importance of parent’s perception and attitude towards medication or treatment, which drives children and adolescents’ medication adherence. Parents’ positive attitudes may influence the child or adolescent’s own attitudes toward medication, which in turn predicted the latter’s adherence [16]. A study conducted by Demidovich et al. [36] stated that parents with high self-efficacy and emotional support were associated with medication refusal. Parents lowered sense of impairment related to the child’s symptom severity and their perception of own ability as sufficient in dealing with the child’s psychiatric disorders, reflects a form or parental resiliency, that resulted in the low perceived need for any medication intervention or medication refusal [36]. Correspondingly, psychiatric disorders are often times subject to stigmatization and may also provide further explanation to parents lowered sense of impairment of the child’s symptom severity and lack of motivation to facilitate medication adherence. Stigma was seen to be positively correlated with perceived costs of psychiatric medication and resistance which then directly discourages the child’s medication adherence [41]. According to Atzori et al. [28], the psychiatrist’s approval of parental request for a weekend drug holiday, as part of accurate treatment planning, have significantly contributed to high medication adherence and progressively demystifies stigma and parent’s negative behavior or attitude towards psychotropic medications.

Several studies have shown that parents’ current psychopathology or a history of hospitalization for a psychiatric disorder was associated with lower medication adherence among their offspring [31,37,38]. However, when the type of psychopathology was taken into consideration, there were mixed findings, such as the inconsistent results found in the association between substance use disorders and child medication adherence [32,36,49]. This may point to the relative influence of other factors ensuing from parental psychopathology. For example, parents with a current psychological disorder may be experiencing active symptoms which leads to an inability to cope with the responsibilities of parenting as well as the complexity of psychiatric treatment regime [32,43,48]. A history of hospitalization for psychological disorders may also indicate more serious psychopathology compared with those with no hospitalization history. Parents with poor mental health may also feel overwhelmed by the additional responsibility of caring for their child who is also facing a mental health condition, and therefore may not be able to closely monitor their child’s medication intake, thus leading to non-adherence among their children [48]. These findings are important, as it shows the importance of providing parents who are also struggling with a mental health condition, with adequate support and skills to ensure the successful medical treatment of their children.

The studies reviewed showed that family living status was not correlated with the child or adolescent’s medication adherence [39,40,43,44,48]. Instead, interpersonal factors which permeated family functionality and relationships were more important. The findings are similar to Timlin et al.’s [16] systematic review of factors contributing to adolescents’ adherence to mental health and psychiatric treatment, which included parental support and family cohesion. Family functioning that are problematic or chaotic with low adaptability, least affectionate and uninvolved in the child’s treatment regime were significantly associated with greater noncompliance with medications [30,34,37,43,44,48,49]. According to Dean et al. [34], even though children tend to have greater responsibility for medication administration as they grow older, it is important for parents to still maintain some parental involvement in medication routines. Woldu et al. [50] further emphasized that parental involvement in the child’s medication regimes should persist in not just younger adolescents but also in those who are distractible and forgetful. In contrast, Timlin et al. [49] reported that the child’s close relationship with the mother is a statistically significant factor in predicting nonadherence. The reason remains unknown and in need of clarification as to whether the children/adolescents’ mothers were opposed to treatments [49].

The definitions of adherence and methods used in assessing medication adherence in the 23 studies that were included in this review varied widely. Self-reported measurements such as questionnaires and interviews with children and adolescents were used to obtain information about medication adherence. Self-reported and subjective report of adherence is a feasible method to obtain information as it is less costly and is correlated with clinical outcomes [71]. However, the self-report methods have their weaknesses, including the inability to ascertain the veracity of the reports, and not being able to control for over- or under-reports of adherence [72]. Self-reported adherence may be even more problematic for children and adolescents as they are vulnerable to responding in a socially desirable manner and young children may have difficulty in understanding the concept and measures of medication adherence. Therefore, corroboration of self-report results are conducted with other-report (e.g., Pogge et al., [47]) and objective measures were also employed, such pill counts (e.g., Atzori et al., [28]), clinical measurements such as serum concentration (e.g., Bernstein et al., [30]), and accessing hospital records (e.g., Ayaz et al., [29]). Therefore, both subjective and objective measures of adherence are important, and should be used in combination to obtain the most rigorous results.

### 4.1. Strengths and Limitations

One major strength of this review lies in the attempt to address the importance of parental characteristics in effects to medical adherence among children and adolescents with psychiatric disorder. There was a range of parental factors addressed in this review inclusive of parent’s sociodemographic or socioeconomic characteristics, parenting style, and family functioning, parent’s social characteristics such as perception, stigma and beliefs, as well as parental psychopathology. Due to relative study of heterogeneity in differing components of parental factors, a meta-analysis was not possible. However, the review process was systematic and all studies included were assessed based on strict eligibility and exclusion criteria to ensure all relevant articles were included in this review. In a similar way, the diversity of medical adherence measures across the articles included in this review were positively seen as a means to reduce or overcome information bias. Nevertheless, it may have also contributed to varying results that prevented causal conclusions from being drawn. All the articles included in this review were limited to English peer-reviewed and published articles in international databases, possibly leaving potential studies published in other languages as well as gray literatures and unpublished articles outside the review. This thus affects the applicability of the review as it confines the generalization of the findings. Consequently, further research is needed to address these constraints and guide the improvement of medication adherence among children and adolescents with psychiatric disorder.

### 4.2. Future Research

The findings of this review will be able to inform future research of the importance of parental factors towards medication adherence. According to the second question addressed in this review, it is shown that parental characteristics, such as parent’s perception and attitude towards medication, parent’s current psychopathology, and parental support or family functioning are significantly associated with medication adherence among children and adolescents with psychiatric disorders. In relation to that, adopting effective positive parenting approaches, such as positive discipline parenting [73] and strength-based parenting [74] that aims to cultivate positive situations, processes, and qualities in children and adolescents, would facilitate the design of tailored strategies to improve adherence in these patients. In addition, this study also found that parental attitudes toward medication was associated with the adherence of their children. Therefore, future studies could investigate methods to improve parental attitudes toward medication. The findings that parents with current psychopathology and a history of hospitalization for psychiatric disorder may indicate the need to further investigate systemic and holistic intervention methods for families dealing with intergenerational psychiatric disorders.

## 5. Conclusions

This study aimed to systematically review studies on parental factors that were associated with medication adherence among children and adolescents with psychiatric disorders. Results from total of 23 studies reviewed showed that medication nonadherence was a highly prevalent and widespread problem among children and adolescents with psychiatric disorders. We found that parents’ socioeconomic backgrounds, family living statuses and functionings, parents’ perceptions and attitudes towards the importance of medication taking in treating psychiatric disorders, and parents’ own mental health statuses were significant parental characteristics associated with their offsprings’ medication adherence. The present study paves the way for future research by allowing active participation of the parents in improving the child’s medication adherence.

## Figures and Tables

**Figure 1 healthcare-11-00501-f001:**
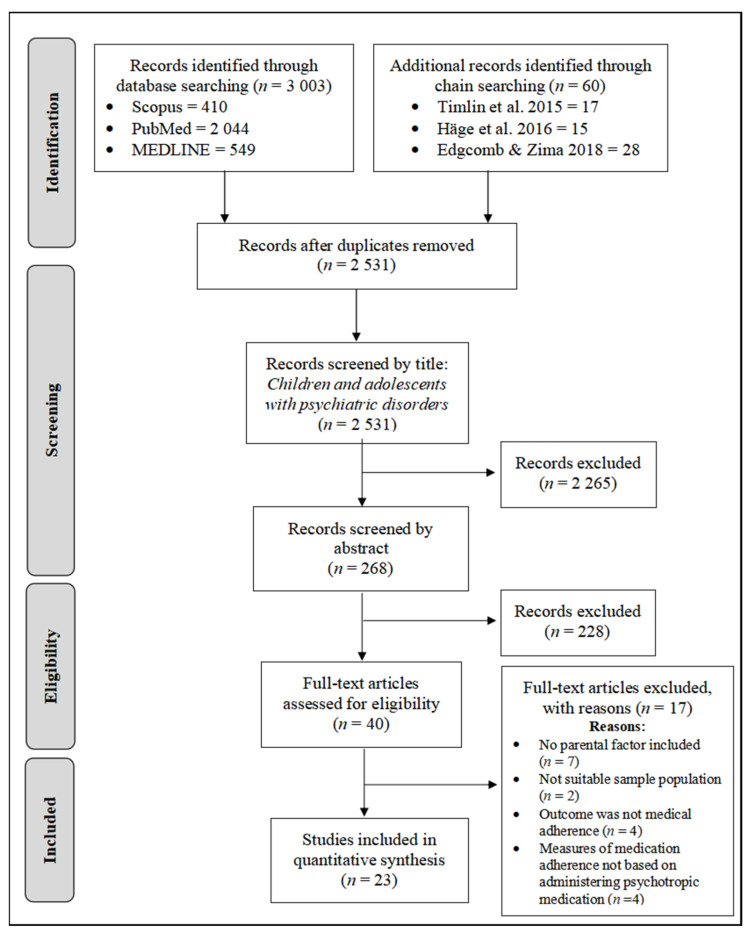
PRISMA flow diagram of search results. Note. PRISMA = Preferred Reporting Items for Systematic Reviews and Meta-Analyses (Edgcomb and Zima [10], Häge et al. [15], and Timlin et al. [16]).

**Table 1 healthcare-11-00501-t001:** Search terms and strategy used in PubMed, Scopus, and MEDLINE (EbscoHost).

[1]	parent–child relations OR parents OR parenting style OR parent * OR parent * style OR parent * approach OR parenting OR positive parent *
[2]	medic * OR medicine OR medication OR medication adherence OR comply OR compliance OR medication compliance
[3]	mental * OR mental illness OR mental disorder OR mental issues OR schizophrenia OR psychiatric disorder OR psych * problem OR mental health OR bipolar disorder OR substance abuse OR psychiatric illness OR depression OR anxiety OR psychotic disorder OR obsessive compulsive disorder OR behavior disorder OR behavioral disorder
[4]	adolescent * OR adolesc * OR teenager OR teen * OR children OR child * OR youth

Note. Truncation technique * for SCOPUS and PubMed.

**Table 2 healthcare-11-00501-t002:** Article characteristics (*n* = 23).

Author	Country	Study Duration	Study Design	Study Setting/Study Location	Sample Population Characteristics		
					Total	Age Group	Excluded Participants
Atzori et al. (2009) [28]	Sardinia, Italy	36 months	Naturalistic Study	Center for Pharmacological Therapies in Children and Adolescent Psychiatry, an outpatient clinic of the Cagliari University Hospital, University of Cagliari	134 children	4–16 years old	(1) Severe side effects (defined as dysphoria=irritability, logorrhea, persistent involuntary movement or over focusing) (*n* = 12) (2) Lack of symptom improvement after at least one week of treatment (*n* = 10) (3) Parental decision, immediately after test doses or during the first 2 weeks of treatment (*n* = 31)
Ayaz et al. (2014) [29]	Turkey	12 months	Not mentioned	Child psychiatry outpatient clinic of Sakarya University Training and Research Hospital	877 children and adolescents	6–18 years old	(1) Families were not reached by phone (*n* = 195) (2) Lack of sufficient data about the treatment efficacy and side effects after each medication switch conducted by clinicians (*n* = 276)
Bernstein et al. (2000) [30]	Not clearly mentioned	8 weeks	Not mentioned	Recruited from a larger study of inpatient adolescents	63 adolescents	12–18 years old	(1) ADHD, conduct disorder, bipolar disorder (or history of bipolar disorder in a first-degree relative), eating disorder, alcohol/drug abuse, mental retardation, or a medical condition that could compromise safe use of tricyclicantidepressants (2) Adolescents taking other psychotropic medications
Burns et al. (2008) [31]	USA	24 months	Prospective study	4 private inpatient psychiatric hospitals in the mid-Atlantic region	85 adolescents	13.3–18.7 years old	(1) No parent or legal guardian resided in the extended metropolitan area (2) Adolescent had mental retardation, was severely neurologically impaired, or was psychotic and judged to be incapable of participating in the interview.
Bushnell et al. (2018) [32]	USA	6 months	Not clearly mentioned	MarketScan Commercial Claims Database (enrollment files, inpatient and outpatient services, dispensed prescriptions)	70,979 children	3–17 years old	(1) Children with diagnostic codes for bipolar disorder, personality disorder, schizophrenia, or autistic disorder in the year before SSRI initiation (2) Children with parents who did not possess 6 months of insurance enrollment following SSRI initiation
Coletti et al. (2005) [33]	Not clearly mentioned	1-month	Not mentioned	Participants were recruited by clinician referral and were receiving outpatient or day treatment services	37 adolescents	12–19 years old	(1) The presence of current psychotic features (2) Possess diagnosis of mental retardation
Dean et al. (2011) [34]	Not mentioned	Immediate	Cross-sectional survey	(1) Child and Youth Mental Health Service (CYMHS) provides child and adolescent tertiary care inpatient unit and three outpatient clinics (2) Participants were recruited via outpatient pharmacy services only	84 children and adolescents	18 years and below	Not mentioned
DelBello et al. (2007) [35]	USA	12 months	Prospective study	Psychiatric Units of Cincinnati Children’s Hospital Medical Center (Inpatient setting)	71 adolescents	12–18 years old	Potential subjects were excluded by a diagnosis of mental retardation (IQ < 70) or a manic or mixed episode resulting entirely from an unstable medical or neurological disorder or acute intoxication or withdrawal from drugs or alcohol, as determined by symptom resolution within 72 h.
Demidovich et al. (2011) [36]	Not mentioned	4 to 6 months	Not mentioned	Community or an outpatient clinic patients recruited through newspaper, radio advertisements and brochures sent to schools and local mental health centers and from program sites affiliated with the University of Pittsburgh Medical Center	96 children	6–11 years old	565 patients excluded due to: (1) Concurrent individual or family participation in a treatment program for disruptive disorders (2) Current psychosis, bipolar disorder, MDD marked by significant vegetative signs, substance abuse, or an eating disorder (3) Suicidality with a plan or homicidality
Drotar et al. (2007) [37]	Not mentioned	20 weeks	Prospective study	Outpatient children and adolescents	107 patients	5–17 years old	(1) A history of intolerance to Li serum concentration Q0.6 mmol/L, DVPX serum concentration Q50 2 g/mL (2) A history of a manic episode with a documented Li serum concentration Q1.0 mmol/L or DVPX serum concentration Q80 2 g/mL (3) The presence of a substance abuse disorder within the previous 6 months (4) Females who were pregnant, at risk of becoming pregnant, or nursing (5) The presence of a clinically significant abnormality on any baseline laboratory measure (thyrotropin blood level, comprehensive metabolic profile, complete blood count, prothrombin time/partial thromboplastin time, urinalysis, urine toxicology screen, and electrocardiogram) and in pulse or blood pressures at study entry (6) Clinical evidence of PDD or mental retardation.
Gearing et al. (2009) [38]	Ontario, Canada	2 years	Retrospective follow-up longitudinal cohort	Psychiatric inpatient hospitals	65 children and adolescents	below 18 years old	Psychotic symptoms were due to substance use or general medical conditions (metabolic or physiologic disorders) during index admission.
Ghaziuddin et al. (1999) [39]	Not mentioned	6 to 8 months	Not mentioned	Adolescent Psychiatry Inpatient Program at a university hospital	71 adolescents	below 18 years old	Not mentioned
Goldstein et al. (2016) [40]	USA	6 months	Naturalistic Study	Child and Adolescent Bipolar Spectrum Services (CABS) clinic at Western Psychiatric Institute and Clinic at the University of Pittsburgh (Specialty Outpatient)	21 adolescents	12 year 0 months– 22 years 11 months	Not mentioned
Harpur et al. (2008) [41]	United Kingdom	Not mentioned	Not mentioned	Participants from UK, US, Canada, Germany, Australia, Israel, Singapore, Republic of Ireland, South Africa, Brazil, and Malaysia recruited through UK ADHD clinics, US (New York) ADHD clinics, ADHD parent support groups, and the internet	123 children	5–18 years old	Not mentioned
Hoza et al. (2000) [42]	USA	14 months	Exploratory study within the context of a well-controlled RCT	Participants recruited from 3 MTA (Multimodal Treatment Study of Children with ADHD) at Pittsburgh and Irvine sites	105 children and adolescents	7–10 years old	Reasons for nonparticipation varied and were not tracked systematically. However, the most common reasons were: (1) Late entry into the protocol (which did not allow time for additional testing before randomization into the main study) (2) Insufficient staffing at the site to allow for extra testing (3) Families declining additional testing beyond what was required for the main study (i.e., nonconsenting)
King et al. (1997) [43]	Not mentioned	6 to 8 months	Naturalistic Study	Adolescent psychiatry inpatient unit	51 adolescents	13–17 years old	(1) Did not meet hospitalization period at baseline evaluation (*n* = 6) (2) Missing follow-up data (*n* = 13)
Moses (2011b) [44]	Not mentioned	Feb 2006 to Aug 2007	Mixed-method	Recruited from a larger project; sample receiving wraparound mental health services in a midsized, Mid-western city	50 adolescents	12–18 years old	Not mentioned
Munson et al. (2010) [45]	USA	Not mentioned	Not mentioned	Participants were recruited through discussions with staff, posters and flyers within outpatient clinic at a large Midwestern university hospital, community mental health settings, and an alternative high school	70 adolescents	12–17 years old	(1) Had not taken psychiatric medication in the past 30 days (2) Had an IQ < 70 (3) Had a PDD, seizure disorder, or an organic brain disorder
Pérez-Garza et al. (2016) [46]	Mexico	3 weeks to 6 months	Not mentioned	Child Psychiatric Hospital in Mexico City	87 adolescents	12–17 years old	Possess active medical comorbidities, drug abuse, and pregnancy
Pogge et al. (2005) [47]	USA	90 days to 18 months	Naturalistic Study	Inpatients from private psychiatric hospital in Westchester County, New York	86 children and adolescents	6–18 years old	Refused to participate (*n* = 12)
Stewart & Baiden (2013) [48]	Ontario, Canada	Immediate	Cross-sectional study	Inpatient treatment in adult psychiatric facilities or units in Ontario	3681 youth	12–18 years old	Not mentioned
Timlin et al. (2014) [49]	Finland	Not mentioned	Not mentioned	Psychiatric inpatient care at the Oulu University Hospital in Finland	72 adolescents	12–17 years old	(1) Subsequent treatment was carried out in a children’s psychiatric ward (*n* = 5), since these wards apply different treatment methods from adolescent psychiatry wards (2) Adolescents did not receive the intended treatment after leaving the acute admission ward (*n* = 4)
Woldu et al. (2011) [50]	Not mentioned	6 to 12 weeks	Not mentioned	Recruited through the Treatment of Resistant Depression in Adolescents (TORDIA) study	190 adolescents	12–18 years old	(1) Possess mania, psychosis, developmental disabilities, substance abuse or dependence, chronic disease (2) Those on a daily medication with psychotropic properties, except for participants who were on a stable dose of a stimulant for ADHD (3) Pregnant or lactating female

**Table 3 healthcare-11-00501-t003:** Association between parental factors and medical adherence.

Reference	Parental Factors	Results	Findings	Assessment Measures
Atzori et al. (2009) [28]	Family living status (living with either natural parents or other figures [adoptive parents or guardians, grandparents, uncles]) Parent preference for a drug holiday for their children Parent motivation to undertake the complex procedures needed to receive MPH as well as the intensive system of care provided by the clinic	100 out of 134 total participants were living with both parents: 41 (66%) were on therapy at 36 months. 27 (84%) were suspended for remission. 32 (82%) were suspended for other reasons.	Adolescents not living with both parents or possessing poor family structure were predictors for continuing medication until end of the 36 months study. ➣ The lack of proper family support, such as living with only one parent, living with either natural parents or other figures such as adoptive parents or guardians, grandparents, uncles, or in foster care, could lead to a scarce availability of educational strategies, making medication the main treatment. Parental preference for a drug holiday on Sunday was allowed for about 75% of the children in the study and it contributed to high treatment adherence. High parental motivation to undertake the complex procedures such as to receive MPH and intensive system of care provided by the clinic explains the high rate of medication persistence. ➣ During each visit, efficacy, tolerability, and safety of the drug were evaluated and discussed with the family; children and adolescents were supported and motivated to continue therapy.	Not mentioned
Ayaz et al. (2014) [29]	Medication efficacy perceived by parents Parent educational levels Biological parents’ togetherness	629 out of 877 children had mothers with low education level. 492 out of 877 children had fathers with low education level. 820 out of 877 children had parents who were together and not divorced. 69 out of 612 children with parents who did not want to give medication discontinued medication.	Insufficient level of efficacy perceived by the parents adversely affected medication persistence, and caused discontinuation of medications. ➣ The willingness of parents to ensure medication persistence is increased when an excess of externalizing behavioral problems in children is perceived. ➣ In the cases of which treatment was discontinued, medication persistence failed because parents did not want to give medication to their children. There is no statistically significant relationship between educational levels of parents and togetherness of parents with medication persistence.	Clinical Global Impressions-Improvement Scale (CGI-I)
Bernstein et al. (2000) [30]	Family cohesion (connected to disengaged) Family adaptability (flexible to rigid) Family functioning or type (extreme, mid-ranged, balanced)	Adolescent ratings of (*n* = 57): ➣ Adaptability (*r* = −0.31, *p* = 0.020) ➣ Cohesion (*r* = −0.27, *p* = 0.045) ➣ Family type (*r* = −0.28, *p* = 0.032) Maternal ratings of (*n* = 56): ➣ Adaptability (*r* = −0.27, *p* = 0.047) ➣ Cohesion (*r* = −0.25, *p* = NS) ➣ Family type (*r* = −0.34, *p* = 0.011)	Adolescent rated low family adaptability (rigidity), low family cohesion (disengagement), and problematic family functioning (extreme family type) were significantly associated with greater noncompliance with medications and therapy appointments. Maternal rated low adaptability and extreme family type were significantly associated with greater noncompliance based on pill counts only. Based on pill counts or on missed appointments, demographic variables, including socioeconomic status, were not significantly associated with noncompliance.	Family Adaptability Cohesion Evaluation Scale II
Burns et al. (2008) [31]	Parental perception of treatment helpfulness Parental lifetime history of psycho-pathology	Parent ratings of pharmacotherapy treatment being helpful: T2 = 14/65 (21.5%) T3 = 17/49 (34.7%) T4 = 15/34 (44.1%) T5 = 12/29 (41.4%)	Parents report on lifetime histories of psychopathology is non-significant and can be considered less relevant to offspring compliance in comparison with current psychopathology. For both forms of therapy at each time point, parents were more likely than adolescents to rate the treatment as helpful. ➣ Parents may be in a better position to perceive positive changes in their adolescents, or they may be more willing to give credit to treatment. ➣ Parents play a crucial role in arranging, transporting, and paying for treatment. When they view treatment as helpful, they may have greater motivation to facilitate treatment compliance.	Child and Adolescent Services Assessment Family History Interview
Bushnell et al. (2018) [32]	Parent’s own medication adherence Parent’s own preventative measure for their own health such as parent well/preventative visits Parent substance use disorder	24,167 out of 70,979 children whose parents have high medication adherence: 63% (15,258) of children had high adherence 10,312 out of 70,979 children whose parents have low medication adherence: 53% (5429) of children had high adherence	Parental medication adherence towards SSRI, statin and antihypertensive at baseline predicted SSRI adherence in children with anxiety disorders. In comparison to parent SSRI adherence, findings were slightly attenuated for parent statin and antihypertensive adherence. ➣ Parent views or behaviors influencing their own adherence, such as taking daily medications and making trips to the pharmacy, may directly or indirectly influence their child’s adherence, if the child shares/models parent views or behaviors. ➣ Statins and antihypertensives are often used as preventative medications. Parents initiating and adhering to these medications may be more likely to partake in healthy behaviors that could influence involvement in their child’s treatment. Parent substance use disorder diagnosis are predictors of low child SSRI adherence while parent well/preventative visit were identified as independent predictors of high child SSRI adherence.	Proportion days covered (PDCs) for SSRI initiation were calculated and dichotomized
Coletti et al. (2005) [33]	Parent’s perception of treatment effectiveness Participants receiving adjunctive family therapy or group therapy	Parents reports: 3 (7.9%) poor adherence 13 (34.2%) optimal adherence 52.6% families receiving family therapy attended all scheduled sessions	Parent effectiveness ratings were high for all treatments (Medication treatment mean = 6.86; Family therapy mean = 7.11; Group therapy mean = 7.43; Individual therapy = 8.12) Parents of children on a mood stabilizer and stimulant regimen reported significantly higher levels of perceived effectiveness (M = 8.57, SD = 1.51) than parents of children on a mood stabilizer and an antidepressant. Non-statistically significant differences were present in adherence patterns for children receiving adjunctive family therapy or group therapy.	Parent reports on 10-point Likert scale
Dean et al. (2011) [34]	Parental involvement in medication routine Family functioning and family lifestyles Shared parenting between more than one household	42 out of 84 children provided information on situations where doses were most likely to be missed: Most frequently cited situation 13 (31%) involved the child staying away from home overnight.	Parental involvement in medication routines was a predictor of good adherence (*p* < 0.05). ➣ Children tend to have greater responsibility for medication administration as they get older, yet the findings reinforce the importance of maintaining some parental involvement in medication routines despite the age of the children. Poor adherence most likely occurs in children where parenting is shared between more than one household, as this may involve disruption to medication routines or parental differences in acceptance of medication use. Poor family functioning that involves chaotic family lifestyles or parent disagreements about diagnosis and management predicted low levels of medication adherence.	Open-ended questions Dichotomous items
DelBello et al. (2007) [35]	Family socio-economic status	Mean score of socio-economic status 3.3 (SD = 1.7)	Low family socioeconomic status (*F* = 4.5, *df* = 1, 70, *p* = 0.04) were associated with nonadherence. ➣ Ø The family SES was higher in medically adhered bipolar adolescents compared with nonadherent bipolar adolescents (*t* = –2.1, *p* = 0.04). ➣ Findings suggest that patients in families of lower socioeconomic status may have limited access to medications and that the quality of care varies among bipolar adolescents of different socioeconomic levels.	Hollingshead scale
Demidovich et al. (2011) [36]	Parental medication acceptability Parental mental health Parental self-efficacy Parent’s emotional support Parenting practices Parental interactions and behavioral management Parental motivation Parental discipline Family relational/interaction Parental demographic factor (Household composition, socio-economic status)	29 (30%) parents of the children declined medication for ADHD.	The decreased parental medication acceptability was related to medication refusers at intake and during study intervention. High parental self-efficacy (*p* = 0.039) and parents’ emotional support (*p* = 0.011) were associated with medication refusal. ➣ This finding reflects the form of parental resiliency that led to a lowered sense of impairment related to the child’s symptom severity, and as a result, parents possess less of a perceived need for a medication intervention. Low parental scores for oppositional defiant symptoms (*p* = 0.033) were significantly associated with medication refusal. Parents with minority status and lower SES were significant correlates with medication refusal at intake and not during the study intervention. ➣ These findings reflect issues of insurance factors, treatment accessibility, feasibility, treatment alliance, and timing of recommendations. There was no significant relationship between parental household composition, parental mental health, parenting practices, parental motivation, parental discipline and family relational or interaction with the youth’s medication adherence.	The Credibility of Treatment Scale (COTS) Patient Health Questionnaire (PHQ) Brief Symptom Inventory Beck Depression Inventory Parental Self-Efficacy Scale Alabama Parenting Questionnaire Parent Perception Inventory University of Rhode Island Change Assessment (URICA) Parent–Child Conflict Tactics Scales and Family Environment Scale-A Family Adaptability and Cohesion Scale.
Drotar et al. (2007) [37]	Maternal educational level Lifetime history of maternal and paternal psychiatric hospitalization Family functioning	Maternal education attainments (*n* = 86) 8 without high school diploma 25 high school graduates without college education 32 some college education 21 degree from 4-year college or more	Based on correlations, maternal educational attainment did not relate to the proportion of DVPX or Li readings of serum concentration in the therapeutic range. Lifetime history of maternal (*r* = −0.31; *p* < 0.01) and paternal (*r* = −0.44; *p* < 0.01) hospitalization for a psychiatric disorder were related to treatment nonadherence for DVPX. Conflictual and problematic family functioning (*r* = −0.26; *p* < 0.05) has been associated with poorer treatment adherence.	Demographic information General Functioning scale of the Family Assessment Device
Gearing et al. (2009) [38]	Family history of depression, bipolar disorder and schizophrenia Family support	Family support: 27 (69%) decreased; 21 (84%) unchanged Family history of depression: 31 (69%) no; 18 (90%) yes Family history of schizophrenia: 14 (25%) no; 2 (25%) yes Family history of Bipolar Disorder: 13 (22%) no; 3 (43%) yes	Several variables were found to be non-significant: age at admission, gender, family history of depression, and decreased family support. Although the authors had anticipated that increased social support would be associated with improved adherence, this hypothesis was not supported by the data. ➣ Ø The population was biased toward including those with relatively high levels of family support reducing the variability in this potential predictor.	Information Update Profile Sheet
Ghaziuddin et al. (1999) [39]	Family living arrangement	Children living with: Biological parents, *n* = 27 (37.7%) Single parents, *n* = 15 (14.4%) Biological + step, *n* = 15 (21.7%) Non-parent, *n* = 14 (20.2%)	The parent–child relationship factors could not be evaluated in this study. ➣ Future research on medication compliance should include direct methods for measurement of patient and parental attitudes toward medication, and measurement of factors which reflect parent–child relationship to obtain significant results.	Follow-up phone interview
Goldstein et al. (2016) [40]	Peer and family influence on medication beliefs and behaviors Family environment Family conflict Living situation (29% lived with both parents)	Mean scores of completed self-reports: Adolescents = 4.7 Parents = 4.3 Physicians = 4.3	Subjective reports from patients, parents, and physicians overestimated adherence as compared with objective data. This is obtained from the poor adherence agreement between self, parent, and physician report. ➣ Youth may change their medication-taking behavior to be in line with provider and/or parent expectations to minimize dissonance or feelings of guilt. Social influences, such as peer and family influence, on medication beliefs and behaviors was not associated with medication adherence. Family environment, family conflict or living situation was not associated with dose omissions.	Family Adaptability and Cohesion Scale-II (FACES-II) Conflict Behavior Questionnaire (CBQ) Social Influence subscale of the Illness Management Survey
Harpur et al. (2008) [41]	Parents’ perceived costs of medication Parents’ perceived benefits of medication Parent stigma Parent dosing flexibility Parent medication-related inconsistency Parental gender Parental marital status	365 parents reported children adherence: 4% parents reported that their children taking medication for < 1 month 12% parents reported that their children taking medication for 1–6 months 12% parents reported that their children taking medication for 6–12 months 20% parents reported that their children taking medication for 1–2 years 24% parents reported that their children taking medication for 2–4 years 28% parents reported that their children taking medication for > 4 years	Parent’s perceived costs of medication were: ➣ positively correlated with resistance; ➣ positively correlated with child stigma; ➣ positively correlated with parent stigma; ➣ negatively correlated with perceived benefits. Parent’s perceived benefits of medication were: o negatively correlated with stigma; o negatively correlated with parental medication related inconsistency. Parental and child stigma were correlated. Interestingly, resistance was most strongly correlated with child stigma, while parental stigma was negatively correlated with perceived inconsistency. Factors such as perceived costs and benefits of medication including child and parent stigma, child’s resistance to taking medication, and parental inconsistency and flexibility are possible markers of adherence medication adherence. There are no significant association between parental gender and marital status with treatment adherence.	➣ Southampton ADHD Medication Behavior and Attitudes (SAMBA) scale
Hoza et al. (2000) [42]	Parental locus of control Parental cognitive errors Parental Self-esteem Parental Self-efficacy Parenting disciplining skills Parents’ attribution of child compliance Parental marital status	105 children with their parents (100 mothers, 57 fathers)	Primary analysis: Both mothers’ (*p* = 0.05) and fathers’ (*p* = 0.07) self-reported use of dysfunctional discipline predicted worse child treatment outcome. Mother: ➣ Mothers with low self-esteem (*p* = 0.06) experience doubts about their parenting ability, making them prone to dysfunctional discipline practices which in turn relate to poorer treatment outcomes through low medication adherence. Father: ➣ Father’s self-efficacy (*p* = 0.06), which is linked to adaptive behaviors such as greater parental responsiveness, greater awareness of community parenting programs and persistence contribute to effective behavior management and also consistent administration of medication intake. Thus, improve treatment outcomes. ➣ Father’s attributions of child compliance (*p* = 0.07), in which they place less blame on their children’s insufficient effort and bad mood, were associated with worse treatment outcomes. There were no significant association between parental marital status, locus of control and cognitive errors towards treatment outcomes.	Internal–External Scale Expanded Attributional Style Questionnaire Cognitive Error Questionnaire Rosenberg Self-Esteem Scale Parenting Sense of Competence Scale Parenting Scale The Interactions Questionnaire
King et al. (1997) [43]	Socio-economic status Family structure Family functioning Parental psychopathology Parental social adjustment	Not mentioned	SES was related to medication follow-through (Fisher’s Exact Test [Monte Carlo Estimate] = 8.32; *p* = 0.051). ➣ Lower and lower-middle SES families, had higher rates of complete follow-through with medication. (82.6%; *X*^2^[2] = 5.18, *p* = 0.075). Family structure or parental residing status was unrelated to treatment follow-through. ➣ This finding may reflect partially of 13 adolescents missing follow-up data, that majority are in single-parent homes (53.8%) or lived either in a home without a parent or in out-of-home placements (38.5%). ➣ Present finding also suggests, despite being a single-parent, yet without interfering psychopathology, single-parents can still overcome structural hurdles to treatment follow-through. The most dysfunctional families and those with the least involved/affectionate father-adolescent relationships had the poorest follow-through with parent guidance/family therapy. ➣ The Med-SOME subgroup, which showed some medication follow-through, had less active/affectionate relationships with their father than did subjects in the Med-COMPLETE subgroup. ➣ One could argue that it is easier to take a pill than it is to travel to an office and discuss personal matters, especially among families with high level of general dysfunction, a depressed parent, or an estranged father-adolescent relationship. Parental psychopathology: ➣ Mothers’ depressive and paranoid symptoms have an impact on a mother’s willingness to accept some responsibility for difficulties and work on family issues, which may result in low treatment follow-through. ➣ Mothers’ high levels of hostility was associated with less medication follow-up. Those in the Med-NONE (*t* [23] = 2.57, *p* < 0.02) and Med-SOME (*t* [23] = 3.23, *p* = 0.004) subgroup had higher maternal hostility scores than did those in the MED-COMPLETE subgroup. ■ Hostility may interfere with some mothers’ ability to trust the physician provider and develop a strong parent physician alliance. There were no follow-through differences related to parental social adjustment, mothers’ anxiety scores, or fathers’ anxiety scores.	Family Assessment Device (FAD) Social Adjustment Inventory for Children and Adolescents (SAICA; parent-adolescent subscales) Symptom Checklist 90-Revised (SCL-90-R) Social Adjustment Scale-Self Report (SAS-SR)
Moses (2011) [44]	Parent’s education Family support Out-of-home living status	Living status: 27 (54%) youths live at foster care, residential facilities, or group homes Perceived family support: (range 1.25–7) Mean (SD) = 5.2 (1.6) score Parent education: ➣ Less committed youths mean (SD) = 12.9 (1.6) years ➣ Committed youths mean (SD) = 14.1 (2.2) years	High perceived family support was significantly correlated with commitment to medication. Clinical or demographic factors, such as the out-of-home living status, and with the exception of parent’s education were not significantly related to medication commitment. Youth classified as less committed to medication had parents who were less educated (*p* < 0.05).	The Multidimensional Scale of Perceived Social Support Sociodemographic Factors
Munson et al. (2010) [45]	Family income	41% reported a yearly family income of less than $50,000	Adolescent’s family income (annual) was significantly related medication adherence (x^2^ = 6.89, *df* = 1, *p* < 0.01). Financial factors or limited resources, such as clinic hours and transportation, remained a significant barrier to care-seeking and continued adherence to treatment all of the time. All adolescents in this study had insurance and did not have access issue. Hence, making co-payments, calling in prescriptions, making trips to the doctor and traveling to the pharmacy, should be taken into consideration, as it can be an added difficulty for financially stressed families.	Not mentioned
Pérez-Garza et al. (2016) [46]	Family belief Pressure/force to take medication	Male adherence ➣ Week 6: 45 (95.7%) ➣ Month 5: 44 (93.6%) Female adherence ➣ Week 6: 15 (68.2%) ➣ Month 5: 14 (70%)	Positive family belief was correlated with adherence among males. Increased pressure or force to take medication was negatively correlated with adherence in females (*r* = −0.55, *p* = 0.009).	Rating of Medication Influence (ROMI)
Pogge et al. (2005) [47]	Parent request of medication discontinuity	36 (43%) participants stopped medication at the request of their parent or a physician	Participants stated that they stopped taking their medication as directed by either their parents or a treatment provider.	Medication adherence interview
Stewart & Baiden (2013) [48]	Residential instability Family/close friends report feeling overwhelmed by youth’s illness Relationship with immediate family members	Residential instability: ➣ Live at Home: 2141 (76.1%) adherent; 674 (23.9%) nonadherent ➣ Temporary Shelter: 633 (73.1%) adherent; 233 (26.9%) nonadherent Family feeling of overwhelmed: ➣ No: 1262 (79.8%) adherent; 320 (20.2%) nonadherent ➣ Yes: 1512 (72%) adherent; 587 (28%) nonadherent Dysfunctional relationship: ➣ No: 1454 (79.7%) adherent; 371 (20.3%) nonadherent ➣ Yes: 1320 (71.1%) adherent; 536 (28.9%) nonadherent	No significant association was found between residential instability, such as living as temporary residence or under a shelter, and medication nonadherence. The proportion of youth that failed to adhere to their medication was greater for those who had a family/close friends that reported feeling overwhelmed by youth’s illness and a disturbed or dysfunctional relationship with immediate family members. ➣ A stable and functional relationship between parents and their youth, while demystifying the stigma associated with some psychiatric conditions, could go a long way to improving medication adherence. ➣ Improving the family dynamics and reducing the family’s sense of overwhelming feelings related to the youth’s mental illness would contribute to medication adherence.	Not mentioned
Timlin et al. (2014) [49]	Family relational functioning Parent’s psychiatric problem Parent’s substance use Parent’s employment	Adolescents close relationship with: 51 (71%) adolescents were closer to mothers; 39 (54%) adolescents were closer to fathers Presence of parent’s psychiatric problems: 5 (7%) mothers; 11 (15%) fathers Presence of parent’s substance use: 8 (11%) mothers; 12 (17%) fathers Parent’s employment: ➣ Mothers: 40 (56%) Full/Part time; 32 (44%) Others ➣ Fathers 45 (62%) Full/Part time; 27 (38%) Others	A statistically significant relationship was found between family relational functioning (*p* = 0.029) and adherence to medicinal treatment. ➣ Adolescents adhered better to medicinal and overall treatment when the family had better relational functioning. A close relationship with the mother is a statistically significant factor in predicting nonadherence. ➣ The mothers’ attitudes towards treatment was not investigated and remains unknown to whether adolescents’ mothers were opposed to treatment. Parents’ psychiatric problems, substance use, and employment did not significantly correlate to adolescents’ adherence to treatment.	Semi-structured Schedule for Affective Disorder and Schizophrenia for School-Aged Children Present and Lifetime (KSADS-PL) interview European Addiction Severity Index (EuropASI) interview Global assessment of relational functioning (GARF)
Woldu et al. (2011) [50]	Family conflict	Not mentioned	Family conflict predicted a failure of child/adolescent remission (Emslie et al., 2010). Parents and youth who were already being attentive to taking one class of medication, will likely adhere to second medication also. ➣ This suggests that parents should monitor adherence to antidepressant medication in depressed youth, particularly closely in younger adolescents, and in those who are distractible and forgetful.	Child Behavior Questionnaire-Parent version (CBQ-P) Child Behavior Questionnaire-Child version (CBQ-A)

**Table 4 healthcare-11-00501-t004:** Description of medication (non)adherence assessment.

Reference	Disorders	Medication	Definition of Adherence or Non-Adherence	Assessment of Adherence or Non-Adherence	% of Adherence	Factors of Adherence or Non-Adherence
Atzori et al. (2009) [28]	ADHD	MPH	Good Compliance Taking at least 80% of the prescribed pills on a monthly basis for at least 8 months per year Still be on treatment with MPH at the end of the study Non-compliance Discontinuation of MPH during the study due to functional remission or other reasons	Checked by the physician for 36 months: counting the unused pills at the end of each month	*n* = 134 children ages 4–16 years 62 (46%) were still on treatment	Presence of associated disorder Younger age Female Not living with both parents
Ayaz et al. (2014) [29]	ADHD	Short-acting MPH Long-acting MPH (OROS MPH) ATX	Medication persistence Determined according to whether or not taking the prescribed medication continued for 12 months after the initiation of treatment Treatment discontinuity A treatment gap >3 months	In a period of 12 months after initiation of treatment: Medication adherence was obtained by parental reports in accordance with the CGI-I scale Clinicians record on medication usage in subject’s file were accessed and data was retrospectively collected through a form prepared by the researcher Phone call interview using a semi-structured interview developed by clinicians, to obtain reasons for medication discontinuation	*n* = 877 children and adolescents ages 6–18 265 (30.2%) continued medication 12 months after initiation of prescription	Younger age Higher hyperactivity/impulsivity symptom severity Use of MPH Addition of another ADHD medication or other psychotropic medication Absence of side effects Perceived medication efficacy
Bernstein et al. (2000) [30]	ODD	Imipramine	Noncompliance Calculated as the number of doses missed divided by the number of doses prescribed, multiplied by 100	Pill count and blood levels for a period of 8 weeks	*n* = 63 adolescents ages 12–18 years *	Diagnosis of ODD Rigidity and disengagement in families
Burns et al. (2008) [31]	Anxiety disorder MDD Dysthymia Bipolar disorder Disruptive behavior disorder GAD CD ODD ADHD Substance, marijuana and alcohol dependence	Antidepressants Mood stabilizers Antipsychotics Stimulants Antiparkinsonian	Noncompliance Participant reported having taken the prescribed medicines rarely, only some of the time, or not at all	Participant fill in the Child and Adolescent Services Assessments at T1, T2, T3, T4 and T5 during the 2-year follow-up	*n* = 85 adolescents ages 13.3–18.7 * years During the course of 2-year follow-up 35 (41.3%) were noncompliant towards medication	Parental perception of treatment as helpful Decreased child psychopathology such as disruptive disorders and affective or anxiety disorders
Bushnell et al. (2018) [32]	Anxiety Disorders inclusive of ➣ PTSD ➣ OCD ➣ Panic Disorder ➣ Social Phobia	SSRIs with Sertraline Fluoxetine Escitalopram Citalopram Benzodiazepine	Adherence Continuation/persistence of a child remained on treatment 6 months after initiation Discontinuation When there was no record of a dispensed prescription 30 days after the previous prescription’s day supply ran out	To capture SSRI adherence, SSRI agents assessed the: 6-month proportion of days covered (PDC) modified medication procession ratio (MPR) 20-day gap in medication coverage between the first and second prescription fills	*n* = 70,979 children ages 3–17 years 40,982 (58%) high adherence	Parental medication adherence
Coletti et al. (2005) [33]	Bipolar disorders (I, II, not otherwise specified)	Mood stabilizer (Lithium or Valproic acid) Antipsychotic Stimulant Antidepressant	Optimal adherence No missed doses Poor adherence Participant missed 10 or more medication	Adolescent psychopharmacologic regimen and adherence measured using a parent questionnaire.	*n* = 37 adolescents ages 12–19 years 13 (34.2%) participants optimal adherence	Longer duration of illness Perceived effectiveness
Dean et al. (2011) [34]	Internalizing disorder (anxiety or mood disorder) Externalizing disorder (ADHD or disruptive behavior disorder) Developmental disorder (PDD, mental retardation)	Stimulants Antipsychotics Antidepressants Sedatives Mood stabilizers Non-psychotropic medication Complementary and alternative medicines	Medication adherence The degree to which the medications taken reflect the prescribed intention.	A 20-min single face-to-face structured interview about medication routine in the past week with parent or child, depending on the primary responsibility for medication taking, was conducted immediately after recruitment using the: Brief Medication Questionnaire (BMQ) self-report measure of adherence Medication History was collected using a standardized template Medication Regimen Complexity Index	*n* = 84 children and adolescents ages ≤ 18 years 32 (38.1%) missing at least one dose 9 (11%) took less than 80% of their prescribed doses 16 (19.0%) took 80–90% 7 (8.3%) took between 90% and 99% of their prescribed doses	High parental involvement Decrease use of complementary medicine Reduced missed doses Developmental diagnosis Increased use of antipsychotics and concomitant non-psychotropic medication
DelBello et al. (2007) [35]	Bipolar disorder	Mood stabilizes (Lithium, Valproic acid, topiramate, lamotrigine) Atypical antipsychotics Antidepressant Psychostimulant	Adherence Full (medication taken >75%) None (medication taken <25%) Partial (medication taken 25–75%)	Medical records of medication use for 12-months post-hospitalization were obtained to assess adherence.	*n* = 71 adolescents ages 12–18 years 25 (35%) full adherence 30 (42%) partial adherence	Absence of ADHD and alcohol use disorder Family socioeconomic status
Demidovich et al. (2011) [36]	ODD or CD with comorbid ADHD	Psychosocial treatment to encourage intake of: Sequential trials of MPH DEX MAS Nonstimulant medications (clonidine or bupropion)	Medication ‘‘acceptors’’ were youth who took a medication at any time during psychosocial treatment. Medication “refusers” were youth with documented referral to psychosocial treatment and they either declined or never showed for an appointment, refused a medication trial when offered by the study psychiatrist, or accepted prescription but never used any medications.	Followed participants through the first 4 weeks to measure adherence via: Clinical Discharge Summary Form Post-treatment KSADS or the Service Assessment for Children and Adolescents (determination of medication history, including current medications at time of intake) Medication Visit Form (current medications, interval history, reported level of compliance, vital signs, Pittsburgh Side Effects Rating Scale and the presence of any other medications taken)	*n* = 96 youths of ADHD cases ages 6–11 years 67 medication acceptors	Parental medication acceptability and intake Decreased parental self-efficacy and emotional support Demographic, family or insurance factors such as high status or socioeconomic status
Drotar et al. (2007) [37]	Bipolar disorder I or II Comorbid diagnosis (anxiety disorder, ADHD, disruptive behavior disorder, substance abuse, eating disorder, learning disorder, motor skill disorder, communication disorder, tics disorder)	Divalproex Sodium (DVPX) Lithium Carbonate (Li)	Nonadherent Range of adherence is 0 to 1 (High scores reflect better adherence)	Both primary and secondary measures were administered for 20 weeks. Primary measures: Serum concentrations measurement Secondary measures: Pill count, patient/parent report of missed pills, and clinical judgment	*n* = 107 adolescents ages 5–17 years 69 (71.8%) children were identified as being adherent to the overall study protocol based on clinical judgement	Maternal and paternal history of hospitalization for psychiatric disorder A greater number of side effects Adaptive family functioning
Gearing et al. (2009) [38]	Psychosis (Primary psychotic disorder or Affective psychotic disorder) Mood disorders with psychotic features	Atypical antipsychotic medication (risperidone, quetiapine, or olanzapine)	Medication adherence Following the physician’s directions If medication type was changed, lowered or discontinued with the agreement and advice of the physician, the participant was considered adherent.	Adherence was measured from the date of discharge (T1) until relapse, identified by readmission to a hospital for a minimum of 3 days for recurrence of psychotic symptoms, or until follow up (T2), minimum 2 years post-discharge, range 24 months to 6.8 years, via parent report in Information Update Profile Sheet.	*n* = 65 children and adolescents ages < 18 years 49 (75%) were adherent to medication	Discharged on concurrent pharmacologic agent for affective symptoms in addition to atypical antipsychotic medication
Ghaziuddin et al. (1999) [39]	Depressive Disorders (MDD, dysthymia, depressive disorder-NOS) Anxiety Disorders (panic disorder with/without agoraphobia, simple phobia, social phobia, OCD, PTSD, GAD, separation anxiety disorder) Disruptive Behavior Disorders (ADHD, conduct disorder, ODD, alcohol and substance use disorders)	Antidepressant (SSRI, tricyclic antidepressants, tetracyclics, bupropion, venlafaxine trazodone) Mood Stabilizers (lithium, valproate, carbamazepine) Neuroleptic Stimulants	Compliance The extent to which the patient’s behavior coincides with the clinical prescription. Noncompliance Discontinuing medication without the recommendation of the treating physician.	Follow-up telephone interview, 6–8 months post-hospitalization on: (1) Name of the medication (2) Whether the medication was discontinued on recommendation of a physician (3) Reason for discontinuation (side-effects, recovery from illness, perceived lack of efficacy) (4) Current living situation	*n* = 71 adolescents ages < 18 years * 24 (33.8%) met the study’s criteria for noncompliance	No conclusion of factors of adherence can be drawn from this study
Goldstein et al. (2016) [40]	Bipolar disorders (I, II, not otherwise specified)	Atypical antipsychotics (Aripiprazole, Risperidone, Quetiapine) Antidepressants (Citalopram, Fluoxetine, Desyrel, Bupropion, Sertraline) Stimulants (DEX, MPH, Lixdexamfetamine) Mood stabilizers (Lithium, Lamotrigine) Others (Levothyroxine, Metformin, Clonazepam, Guanfacine)	Poor adherence Less than 80% of prescribed medication taken or gaps in medication of at least 7 days	Both objective and subjective methods were administered for 6-months. Objective methods: electronic pillbox (MedTracker ratings: adherent dose, wrong-time dose, wrong-day dose, dose omission)Subjective methods rated on a 1–5 rating scale by adolescents, parents and prescribing physicians: (1) almost never missed (<10% of the time) occasionally (2) missed (10%–25% of the time) (3) often missed (25%–50% of the time) (4) missed most of the time (50%–80% of the time) (5) almost always missed (>80% of the time)	*n* = 21 adolescents ages 12 year 0 months–22 years 11 months Objective method: 5 (23.8%) subjects met the accepted criteria for adherence Subjective method: *	Low daily doses Lower weight, Dose timing, More temporal proximity to medication management appointment Lower self-reported cognitive difficulties with adhering to treatment Lower overall illness severity
Harpur et al. (2008) [41]	ADHD	Sustained release MPH (SR-MPH) Short acting MPH (SH-MPH) MPH MAS ATX Dextroamphetamine (DEX)	Not clearly defined	Self-reported hard paper copies/online of the Southampton ADHD Medication Behavior and Attitudes scale (SAMBA; information from the parent and child about medication routine in the past 3 months and factors associated with adherence)	*n* = 123 children ages 5–18 * years	Perceived costs and benefits of medication Child and parent stigma Child’s resistance to taking medication Parental inconsistency and flexibility
Hoza et al. (2000) [42]	ADHD	MTA Treatment in 4 groups: Medication Management (MedMgt) Behavioral Treatment (Beh) The combination of MedMgt and Beh (Comb) Community Care (CC)	Non-compliance Perceived when less endorsement of insufficient effort or bad mood of child	Self-reported measure by parent and teachers for a period of 14-months through the: The Interactions Questionnaire. (INTX) designed to assess parents’ attributions for their children’s compliance or noncompliance in hypothetical interaction situations SNAP-IV rating scale	*n* = 105 children and adolescentsages 7–10 * years	Parenting styles Parental self-esteem Parental efficacy Parent attributions or cognitions about their children and parenting Parent locus of control
King et al. (1997) [43]	Mood disorders (MDD, Bipolar disorder, Depressive disorder not otherwise specified, Dysthymia) Behavioral and substance use (Alcohol use disorder, other substance use disorder, CD, ADHD, ODD) Others (Eating disorders, Social phobia, GAD, Separation anxiety disorder, PTSD)	Psychoactive medication	Adherence coded at: None (0 or 1 contact) Some (>1 contact but discontinued without professional recommendation) Complete (information about the extent of follow-through was not reported)	Structured telephone interviews for a follow-up period of 6-months	*n* = 51 adolescents ages 13–17 years was recommended psychoactive medication 7 (13.7%) some follow-through 34 (66.7%) complete follow-through	Family functioning More involved/affectionate father-adolescent relationships Parent psychopathology Parent hostility
Moses (2011b) [44]	Affective disorder (depression, anxiety including OCD, mood disorder NOS, bipolar disorder) Disruptive behavior disorder (ADHD, Conduct disorder, ODD, Disruptive behavior disorder NOS)	Antipsychotic Antidepressant Stimulant Mood stabilizer or anti-convulsant	“Committed” continue taking medication to avoid unwanted psychological states or behaviors assumed will manifest if they do not take their medication “Less committed” would likely stop taking psychotropic drugs if there were no external pressures proactive’ non-commitment category communicated clear intention and certainty to not take medication	Qualitative semi-structured interview immediately after recruitment via: Medication Commitment Medication Experience Perceived Treatment Choice or Coercion	*n* = 50 adolescents ages 12–18 years 1/3 sample or 19 (38%) was classified as ‘‘committed’’	Taking antipsychotic medication Greater perceived family support Lack of perceived coercion to take the medication
Munson et al. (2010) [45]	Mood disorders	Antidepressant Antipsychotics Psycho-stimulants	Fully adherent Youth reported they adhere to their mental health appointments and medications all of the time	Self-reported adherence was measured by a single indicator ‘‘I [My child] take my medication just as it is prescribed.’’ with response options: (1) Not at all (2) Sometimes (3) Usually (4) (4) All the time	*n* = 70 adolescents ages 12–17 years 63% reported taking their medications as prescribed all of the time	Family income Overall attitudes towards mental health services
Pérez-Garza et al. (2016) [46]	Schizophrenia or Schizophreniform disorder	Antipsychotic medication (risperidone)	Medication adherence defined in a dichotomous variable (the constituents of Yes/No were not mentioned), assessing the patient’s capability to follow the medication instructions	Self-reported Rating of Medication Influences (ROMI) administered throughout a 6-month follow-up. The ROMI contained: compliance subscales (prevention, influence of others, and medication affinity) non-compliance subscales (denial/dysphoria, logistical problems, rejection of label, family influence, and negative therapeutic alliance)	*n* = 87 adolescents ages 12–17 years *	Females in this study showed an inconsistent adherence. Males factors of adherence are: Perceived benefits of medication Relationship with clinician Positive family belief Relapse prevention Pressure or force Fear of rehospitalization
Pogge et al. (2005) [47]	Depressive Disorder Mood disorders NOS Substance Abuse Disorder Disruptive Behavior Disorder Psychotic disorder Anxiety disorder Bipolar disorder ADHD Eating disorder	Olanzapine Risperidone	Adherence Taking medication as directed Noncompliance Discontinuation of medication (patient’s decision) Discontinued group (Discontinuation of medication directed by parents or another physician)	Structured interview with patients, who are contacted after an average of 10 months discharge from hospital, about each medication prescribed, experience of various side effects, patients’ beliefs about medication, as well as drug and alcohol use.	*n* = 86 children and adolescents ages 6–18 years 38 (45.2%) adherent	Positive beliefs about medication effectiveness Low substance abuse Low weight gain
Stewart & Baiden (2013) [48]	Not stated	Not stated	Nonadherence Taking less than 80% of their prescribed medications during the month prior to admission or refused to take some or all of their prescribed medications in the last 3 days preceding the assessment	Resident Assessment Instrument for Mental Health (RAI-MH) completed by trained clinical hospital staff using all sources of information available including interviewing patients, family, friends, clinical chart notes, clinical observation, etc.	*n* = 3681 youths ages 12–18 years 2772 (75.3%) were adherent	Side effects of psychotropic medication Age Psychiatric admissions Insights into mental health Family functioning Reduce use of tobacco and cannabis Low depressive and positive symptoms
Timlin et al. (2014) [49]	Substance use disorder Anxiety disorders Affective disorders Conduct or ODD or ADHD Psychotic disorders	Sedative/Anksiol Neuroleptic Antidepressive Lithium Stimulant	Full adherence Patient takes the medication at least 75 % of the total number of times recommended by the physician in the absence of external pressure If possible; medically and psychiatrically asymptomatic and stable upon discharge Non-adherence Discontinues medication without medical advice or fails to take medication as prescribed	Hospital records or case notes through a clinical follow-up project of STUDY-70	*n* = 72 adolescents ages 12–17 years 56 (78%) adhered to medicinal treatment	Family or social network relational functioning Use of special services at school Adolescents that do not exhibit in self-mutilative behavior and involuntary treatment
Woldu et al. (2011) [50]	MDD	SSRI Fluoxetine Paroxetine (Citalopram substitute for paroxetine after Food and Drug Administration warning) Venlafaxine	Nonadherence LDR = A twofold or greater variation in the dose-adjusted concentration of drug plus metabolites (level/dose ratio [LDR]) CPC = > 30% of the prescribed pills remaining	The adherence rate was measured after 6 and 12 weeks of treatment via: Ratio of the plasma concentration of drug metabolite divided by dose (level–dose ratio [LDR]) Clinician Pills Count (CPC) (pills ratio should be ≥70% to be rated as adherent)	*n* = 190 adolescents ages 12–18 years CPC—92 out of 181 adolescents were adherent (50.8%) LDRI—121 out of 190 adolescents were adherent (64%)	Younger age

Note: ADHD = Attention Deficit Hyperactivity Disorder; ODD = Oppositional Defiant Disorder; PTSD = Post-Traumatic Stress Disorder; OCD = Obsessive-Compulsive Disorder; MDD = Major Depressive Disorder; CD = Conduct Disorder; GAD = Generalized Anxiety Disorder; NOS = Not Otherwise Specified; MPH = Methylphenidate; ATX = Atomoxetine; DEX = Dextroamphetamine; MAS = Mixed Amphetamine Salts; SSRIs = Selective Serotonin Reuptake Inhibitors; * no (further) information on % of adherence provided in publication.

## Data Availability

Data can be obtained from the corresponding author upon reasonable request.

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
