# Peer review of "Parental Factors Associated with Child or Adolescent Medication Adherence: A Systematic Review"

_healthcare, 2023, doi:10.3390/healthcare11040501_

Round 1
Reviewer 1 Report
This paper presents a systematic review on an important topic: Parental factors associated with child or adolescent medical adherence. It makes a good contribution to the literature. Overall, I found the topic very interesting and the writing quite strong. The subject is relevant, the aims are clear, and you have chosen an appropriate research methodology, and I believe you will contribute to our understanding of this important aspect. With that said, it is in the spirit of strengthening the manuscript that I offer the following recommendations:
Abstract: Delete “Introduction”. Also instead of: “adolescents presenting with psychiatric disorders”, it would be better “adolescents with psychiatric disorders”
Introduction: p. 3 The results of previous systematic results should be succinctly presented
Results: The tables 4 and 5 format is poor. It should be improved.
Something is missing in p. 31, line, 12 “terms often used interchangeably (cf. Table 5) Majority of the studies (n = 20)”
Conclusions: Future research areas should be given more thought and expanded upon.
Author Response
Response to Reviewer 1 Comment
This paper presents a systematic review on an important topic: Parental factors associated with child or adolescent medical adherence. It makes a good contribution to the literature. Overall, I found the topic very interesting and the writing quite strong. The subject is relevant, the aims are clear, and you have chosen an appropriate research methodology, and I believe you will contribute to our understanding of this important aspect. With that said, it is in the spirit of strengthening the manuscript that I offer the following recommendations:
Point 1: Abstract: Delete “Introduction”. Also instead of: “adolescents presenting with psychiatric disorders”, it would be better “adolescents with psychiatric disorders”
Response 1: Thank you for your recommendation. We have made the changes accordingly throughout the manuscript.
Point 2: Introduction: p. 3 The results of previous systematic results should be succinctly presented.
Response 2: A few systematic reviews have been conducted in the area of medical adherence among children and adolescent with psychiatric disorder. We have added a paragraph on the summary of findings from the past systematic reviews in the introduction section as suggested.
(Introduction, p. 3) There are few reviews that have reported the role of parents as one of the factors associated with medication adherence among children and adolescents with presenting psychiatric disorder [20, 30, 62]. Both Edgcomb and Zima [20] and Häge et al. [30] investigated predictors of medication adherence only, while Timlin et al. [62] reviewed factors associated with adolescents’ adherence to both medication and non-pharmacological treatments in mental health. A total of 60 studies were reviewed in Edgcomb and Zima [20], Häge et al. [30] and Timlin et al. [62]. The reviews concluded that the range of medication nonadherence was wide, be-tween 6% and 62%, and was considered a common problem in mental health care among children and adolescents with a psychiatric disorder. Factors such as illness severity, comorbidity burden or underlying diagnosis, substance use, and attention-deficit/hyperactivity disorder, age, sex, interpersonal care processes and the adolescent’s own beliefs towards treatment emerged as significant predictors of adherence. With regards to parental factors, the findings from these reviews suggests that positive attitudes or the level of support obtained from family members were associated with higher adherence among children and adolescents with psychiatric disorders [20, 30, 62]. Nevertheless, Timlin et al. [62] pointed out the fact that it is challenging to ensure adolescents' medication adherence to prescribed treatment or medication regimens, as they are transitioning into adulthood and tend to become more independent of their parents. However, these re-views did not provide a clear synthesis of literature that highlights specific components of the parental factors associated with child/adolescent medication adherence. Häge et al. [30] also emphasized the need for future research that involves familial factors associated with medication adherence among children and adolescent with psychiatric disorders. Thus, the purpose of this review is to evaluate the peer-reviewed literature addressing specific aspects of parental factors that are positively or negatively associated with medication adherence among children and adolescents diagnosed with psychiatric disorders.
Point 3: Results: The tables 4 and 5 format is poor. It should be improved.
Response 3: The format of Table 4 has been simplified by deleting the last column. The format of Table 5 has been improved by removing the last column and essential data such as % of (non)adherence to the medication has been added. Since Table 3 has been moved to Supplementary Table S1, hence Table 4 has been renamed as Table 3 and Table 5 has been renamed as Table 4.
Point 4: Something is missing in p. 31, line, 12 “terms often used interchangeably (cf. Table 5) Majority of the studies (n = 20)”
Response 4: Thank you for pointing this out. We have added a full stop to the sentence.
(Results, p. 22) In studies conducted by Ayaz et al. [4], Bushnell et al. [9], and Demidovich et al. [17], “medication acceptors/ medication persistence” or “medication refusers/ discontinuation” were some of the synonymic terms used to address adherence, however the term “compliance/ adherence” or “noncompliance/ nonadherence” were among the common terms often used interchangeably (cf. Table 4). Majority of the studies (n = 20) included in this review mainly attempted to investigate medication adherence among children or adolescent with psychiatric disorders.
Point 5: Conclusions: Future research areas should be given more thought and expanded upon.
Response 5: Future research areas such as strength-based parenting and positive discipline parenting as part of parental factors associated with child/adolescent medication adherence have been added in a separate section namely “Future Research” and changes in the conclusion was also made.
(Future Research, p. 32) The findings of this review will be able to inform future research of the importance of parental factors towards medication adherence. According to the second question ad-dressed in this review, it is shown that parental characteristics, such as parent’s perception and attitude towards medication, parent’s current psychopathology and parental support or family functioning are significantly associated with medication adherence among children and adolescents with psychiatric disorders. In relation to that, adopting effective positive parenting approaches, such as positive discipline parenting [47] and strength-based parenting [68] that aims to cultivate positive situations, processes, and qualities in children and adolescents, would facilitate the design of tailored strategies to improve adherence in these patients. In addition, this study also found that parental attitudes toward medication was associated with the adherence of their children. Therefore, future studies could investigate methods to improve parental attitudes toward medication. The findings that parents with current psychopathology and a history of hospitalization for psychiatric disorder may indicate the need to further investigate systemic and holistic intervention methods for families dealing with intergenerational psychiatric disorders.
(Conclusion, p. 32) This study aimed to systematically review studies on parental factors that were associated with medication adherence among children and adolescents with psychiatric disorders. Results from total of 23 studies reviewed showed that medication nonadherence was a highly prevalent and widespread problem among children and adolescent with psychiatric disorders. We found that parent’s socioeconomic background, family living status and functioning, parent’s perception and attitude towards the importance of medication taking in treating psychiatric disorders, and parent’s mental health status were significant parental characteristics associated with their offspring’s medication adherence. The pre-sent study paves the way for future research by allowing active participation of the parents in improving the child’s medication adherence.
Reviewer 2 Report
Thank you for giving me the opportunity to review the article entitled “Parental factors associated with child or adolescent medical adherence: A systematic review”
Methods
In Figure 1. PRISMA flow diagram, I am not clear about the selection of diagnostic criteria, according to DMS-V. All the articles that were included had to have a diagnosis according to the DSM-V classification? If so, I find it strange that the authors have removed so few articles, since most of the articles tend to use tests or tools for validating conditions.
Why did the authors not indicate the adherence rate in the results table? “Table 2. Article characteristics (n=23)”. According to the authors, it is an inclusion criterion and an objective of the study.
I suggest moved to supplementary files “Table 3. Quality assessment of the studies included based on STROBE”
The authors do not adhere to the journal's standards. References are enclosed in square brackets. There are very serious formatting errors.
Results
Table 4. Association between parental factors and medical adherence. This table is very confusing, the information is not well understood or summarized. In addition, essential data such as % adherence or non-adherence to the medication are not indicated. For Table 5 similar comments.
I think that this work, although it has a good search base, lacks understandable results.
Author Response
Response to Reviewer 2 Comment
Point 1: Methods: In Figure 1. PRISMA flow diagram, I am not clear about the selection of diagnostic criteria, according to DMS-V. All the articles that were included had to have a diagnosis according to the DSM-V classification? If so, I find it strange that the authors have removed so few articles, since most of the articles tend to use tests or tools for validating conditions.
Response 1: Thank you for pointing this out. We have changed the criteria into children and adolescent with psychiatric disorders in Figure 1.
Point 2: Methods: Why did the authors not indicate the adherence rate in the results table? “Table 2. Article characteristics (n=23)”. According to the authors, it is an inclusion criterion and an objective of the study.
Response 2: The % of adherence has been added in Table 5.
Point 3: Methods: I suggest moved to supplementary files “Table 3. Quality assessment of the studies included based on STROBE”
Response 3: Thank you for your suggestion. We have moved Table 3 to supplementary files.
Point 4: Methods: The authors do not adhere to the journal's standards. References are enclosed in square brackets. There are very serious formatting errors.
Response 4: We apologize and thank you for pointing this out. The changes according to the journal’s reference or formatting standard have been made.
Point 5: Results: Table 4. Association between parental factors and medical adherence. This table is very confusing, the information is not well understood or summarized. In addition, essential data such as % adherence or non-adherence to the medication are not indicated. For Table 5 similar comments.
Response 5: The format of Table 4 has been simplified by deleting the last column. The format of Table 5 has been improved by removing the last column and essential data such as % of (non)adherence to the medication has been added. Since Table 3 has been moved to Supplementary Table S1, hence Table 4 has been renamed as Table 3 and Table 5 has been renamed as Table 4.
Reviewer 3 Report
Dear Authors,
Overall, the article was well writtenand structured. However, there are notable gramatic mistakes and references are not arranged in order.
Please check my comments in the PDF file with yellow highlighted tect boxes from the abstract.
Thank you

Author Response
Response to Reviewer 3 Comment
Overall, the article was well writtenand structured. However, there are notable gramatic mistakes and references are not arranged in order.
Please check my comments in the PDF file with yellow highlighted tect boxes from the abstract.
Point 1: Please check the grammatic mistakes. You used the present and past tense in the same sentence. Example: “is” and parental factors “were”. It must be revised.
Response 1: Thank you for your recommendation. The sentence has been revised, as follows:
(Abstract, p. 1) The purpose of this study is to systematically review studies addressing specific aspects of parental factors that are positively or negatively associated with medication adherence among children and adolescents with psychiatric disorders.
Point 2: “A systematic literature search of English language publications, from inception through December 2021 that involved the PubMed, Scopus and MEDLINE databases was undertaken.” This sentence is very hard to understand. Please write in simple manner.
Response 2: The sentence has been simplified the sentence, as follows:
(Abstract, p. 1) A systematic literature search of English language publications, from inception through December 2021 was conducted from PubMed, Scopus and MEDLINE databases.
Point 3: “A total of 23 studies (n = 77,188) met inclusion criteria. Nonadherence rates ranged between 8% to 69%.” Suggested to remove the “n” in ().
Response 3: The “n” has been removed from the brackets. We have instead used the term (77,188 participants).
Point 4: Please remove the subheadings in introduction part.
Response 4: We have removed all subheadings from the introduction section.
Point 5: Reference numbers must be cited in order, it must be started from no. [1]. Authors should arrange the references one by one in order.
Response 5: Thank you for pointing this out. The changes have been made accordingly throughout the manuscript.
Point 6: “Globally an estimated 13% of adolescents aged 10-19 years old experienced mental disorders that are often times unrecognized and untreated.” What is the age group for adolescents? Please check and provide the relevant reference if it included 10 years.
Response 6: We apologize for leaving this detail out in the manuscript. We have used to age-bracket of 10 to 19 years old to define adolescence, according to WHO definitions. The refencence has been added in the manuscript, as follows:
(Introduction, p. 1) Globally an estimated 13% of adolescents aged 10-19 years old [5] experienced mental disorders that are often times unrecognized and untreated.
Point 7: “There are many determinants of medication adherence that were reported and categorized in various ways.” Recommended to add the references of the articles reported determinants of medication adherence.
Response 7: Thank you for your suggestion. Articles on determinants of medication adherence has been cited and added in the reference list.
Reviewer 4 Report
A nice read!
Some grammatical mistakes need to be fixed.
Author Response
Response to Reviewer 4 Comment
Point 1: In my opinion your manuscript can be accepted for publication. Only grammar still needs attention (e.g. see the suggestions of Reviewer 3).
Response 1: Thank you. We have fixed the grammatical mistakes as suggested by Reviewer 3.